# Application of Fractal Dimension and Texture Analysis to Evaluate the Effectiveness of Treatment of a Venous Lake in the Oral Mucosa Using a 980 nm Diode Laser—A Preliminary Study

**DOI:** 10.3390/ma14154140

**Published:** 2021-07-25

**Authors:** Mateusz Trafalski, Marcin Kozakiewicz, Kamil Jurczyszyn

**Affiliations:** 1Department of Dental Surgery, Wroclaw Medical University, Krakowska 26, 50-425 Wroclaw, Poland; mateusz.trafalski@umed.wroc.pl; 2Department of Maxillofacial Surgery, Medical University of Lodz, 113 S. Żeromski Street, 90-549 Lodz, Poland; marcin.kozakiewicz@umed.lodz.pl

**Keywords:** venous lake (VL), diode laser 980 nm, texture analysis, fractal dimension analysis, laser surgery

## Abstract

Venous lake (VL) is a common lesion resulting from the enlargement of thin-walled veins surrounded by a thick wall of fibrous tissue. Photocoagulation is becoming one of the basic methods for treating small vascular lesions. The aim of this study was an application of texture analysis (TA) and fractal dimension analysis (FDA) to evaluate VL treatment. Twenty-three VLs were included in the study. We used a 980 nm diode laser, 6 W, 100 ms pulse mode with a 50% duty cycle. The total dose of energy was in a range from 80 J to 600 J. We used the difference of intensity algorithm for FDA and microcontrast and a co-occurrence matrix for TA. A complete therapeutic effect was achieved in 83%, and in 9%, scar formation was observed after three months. No healing was observed in 4%, and there was partial healing in 4%. No differences in FD were observed between the lesions after three months and the healthy mucosa. The fractal dimension and microcontrast of a vascular lesion are mutually coupled. FDA and TA is a useful and objective method of assessing treatment effects for venous lakes. The non-contact mode of the 980 nm diode laser is an effective and safe method for treating a VL.

## 1. Introduction

A venous lake (VL) is a common lesion resulting from the enlargement of thin-walled veins surrounded by a thick wall of fibrous tissue [1]. In 1956, Bean and Walsh described these lesions as being soft, easily compressible and arising in sun-exposed areas (ears, face, hands, lips), especially in older people [2]. They are also present on the oral mucosa, the red zone of the lip and the ventral surface of the tongue. In the oral cavity, they appear as non-painful, soft, non-bloody, well-demarcated, dark blue or purple papules, 2–10 mm in diameter and showing a positive diaphoresis test [3]. Another name, senile haemangioma or lip varices, can also be found in the literature [4]. The mechanism of their formation is not fully explained, but age is the most likely factor, as they are most common in older people over 50 years of age. If they do not cause aesthetic problems or are not a source of bleeding, no treatment is necessary. However, they can have a negative impact on quality of life, mainly due to aesthetic concerns or fear of malignancy, which lead patients to seek specialist help. Contemporary methods of treatment of these lesions include surgical excision, cryosurgery using liquid nitrogen, electrocoagulation, coagulation with infrared light, photocoagulation and sclerotherapy [5,6,7,8]. If the location of these lesions includes aesthetically sensitive areas, this poses additional requirements for the therapeutic option. This involves the selection of the least invasive method, which will not leave scars or discolouration and has a range not exceeding the borders of pathological changes for the given case. The absorption spectrum for the haemoglobin contained in the blood found in the lumen of vascular lesions covers two spectral regions, 410–600 nm and 800 nm–1100 nm [9]. By selecting the right laser device, such lesions can be selectively treated. Therefore, photocoagulation is becoming one of the basic methods for treating small vascular lesions in the oral area.

Clinical evaluation of the efficacy of treatment of superficial VL is relatively simple. Their characteristic round or oval shape, clear demarcation from the substrate and dark colour make monitoring the progress of treatment easier. Therefore, the response to treatment, or lack thereof, is clearly observable. However, macroscopically, it is difficult to assess whether the cured area is similar in appearance to the normal mucosa. By using texture analysis (TA) and fractal dimension analysis (FDA) to evaluate the efficacy of VL treatment, the results of our study became more objective.

Fractal dimension analysis (FDA) is used in the estimation of complex, irregular shapes. In medicine, FDA is useful during analyses of positron emission tomography, radiographic images, computed tomography and magnetic resonance images [10,11,12,13]. Euclidean geometry may fail in the analysis of complicated shapes. Using classical Euclidean geometry on a daily basis, we are used to the fact that the number of dimensions that an object possesses is an integer. Thus, a point has zero dimensions; a segment has one dimension, which is its length; an item on a plane has two dimensions, length and width; and a 3D item has height, width and length. However, fractals are shapes that go beyond the rules of classical geometry. Their dimensions are not integers. Another surprising feature of fractals is their self-similarity. This is a feature that manifests itself in the fact that fractals can be enlarged without limit and that subsequent details of their structure are similar to the initial shape. In everyday life, we are confronted with numerous natural creations that can be roughly described as fractals, e.g., networks of blood vessels or nerves. There are a number of mathematical methods for calculating fractal dimension—in our study, we used a modified box counting method that allowed analysis of grayscale images.

Another mathematical image analysis that we used in our study was texture analysis (TA). Every digital image consists of pixels. Each pixel is characterised by two features: coordinates and colour/brightness. Digital colour images produced by typical cameras have a colour depth of 24 bits. This means that three components (responsible for red, green and blue) are needed to describe a coloured pixel. Each element can be in one of 256 (2^8^) brightness levels. The pixels form a fine image structure called a texture. A texture is a set of repeating graphic patterns characterised by brightness, entropy, smoothness, uniformity, roughness, granularity, randomness or linearity [14]. TA offers the possibility of surface analysis. Texture analysis is widely applied to magnetic resonance images, computed tomography and X-ray images [15,16,17,18].

The first aim of the study was to investigate the efficacy of oral mucous venous lake lesion treatment, using a 980 nm diode laser. The second goal of the study was the application of texture and fractal dimension analysis in the assessment of the effectiveness of treatment of these lesions. We did not find more studies in the literature about the application of fractal dimension and texture analysis in cases of treatment evaluations of venous lake lesions.

## 2. Materials and Methods

### 2.1. Patients and Lesions

Twenty-three VL lesions in 23 patients (12 women and 11 men) were included in the study. The mean age was 66 years (SD = 14). The youngest patient was 36, and the oldest was 87 years old. The mean age for women was 66, and for men, it was 64 years. The most common location of VL was the lip, and the rarest location was the tongue. The number of cases for the lip was 11; for the corner of the mouth, it was 6; for the cheek, it was 4; and for the tongue, it was 2. Patients suffering from cancer or undergoing radiotherapy or chemotherapy for head and neck cancer, undergoing immunosuppression or with congenital tumours and venous malformation were excluded from the project. The project was approved by the Bioethics Committee at the Medical University of Wrocław, approval number KB—805/2019 (2 December 2019) and regulatory number CWN UMED: BW-110. The diagnosis was based on a thorough history and examination. The circumstances and the time elapsed since the appearance of the lesion, the accompanying symptoms, the growth dynamics, the appearance of the lesion and the symptom of the diascopy were all taken into account. The patient underwent a physical examination and anamnesis before the treatment with follow-up visits at 7, 14 and 90 days after the treatment. During each visit, the lesion was measured with a graduated periodontal probe, and photographic documentation was taken. The complete absence of a lesion or a slightly lighter mucosa (scar) at the site of the treatment were considered criteria for remission. If there was no healing 3 months after the procedure, a repeat procedure was performed with the same frequency of follow-up visits.

### 2.2. Laser Procedure

Before the procedure, the mucosa was superficially anaesthetised with 10% lidocaine (Lidocain-Egis solution 10%, Delfarma, Łódź, Poland), using an occlusive dressing (soaked surgical gauze) placed on the mucosa for 5 min. The operator as well as the patient wore safety goggles during the procedure. The procedure was performed using a 980 nm diode laser (Lasotronix Smart M, Piaseczno, Poland). The optical fibre diameter was 400 µm, power 6 W and illumination in 100 ms pulse mode with a 50% duty cycle. The total dose of energy emitted during the treatment depended on the size of the lesion and ranged from 80 J to 600 J. The irradiation was carried out in non-contact mode through a 1 mm thick transparent microscope slide. The slide was applied to the lesion with slight pressure in order to reduce its vertical dimension and thus, better penetrate the laser beam deep into the tissue. The transparency of the glass was 90%. The Power and Energy Meter PM100USB, Thorlabs Inc, Newton, NJ, United States was used to assess this parameter. Therefore, 90% of the energy emitted by the device was transferred to the tissue, and the remaining 10% was scattered and reflected by the glass. The tip of the fibre optic was then applied at right angles to the lesion surface, and the laser device was activated. The procedure was carried out until slight convergence was achieved at the exposure site, and then, the tip was moved alongside. Once slight convergence of the entire lesion surface was achieved, the procedure was completed. Perioperative pain was determined using the NRS (numerical rating scale), which contained 10 levels of pain severity, where 1 was minimal pain and 10 was unbearable pain. Intraoperative pain intensity averaged 4.76. Only in three cases was the pain so severe that infiltration anaesthesia had to be performed using 4% articaine with norepinephrine 1:200,000 (Septodont, Saint-Maur-des-Fossés, France). The lesions were left to heal on their own. It was recommended to avoid irritation of the treated area and not to eat hot and spicy food. No patient required analgesics, and in all cases, the healing progressed without complications.

### 2.3. Taking Images

All images were taken with the camera of a Samsung S9 mobile phone (Samsung Electronics, Seoul, South Korea) in RAW format to reduce the effects of sharpening and denoising algorithms. All images were taken from the same distance (45 cm), and the optical axis of the camera was perpendicular to the lesion surface. The focus point was locked at 45 cm for repeatability. Depending on the lesion area, a region of interest (ROI) of 50 × 50, 100 × 100 or 150 × 150 pixels was determined. All ROI images were converted to 8-bit greyscale bitmaps.

All graphic operations were performed using GIMP version 2.10.24 (GNU Image Manipulation Program—www.gimp.org (accessed on 3 May 2021), free and open source license).

### 2.4. Measurement of Lesion Surfaces

Imagine J software, version 1.53e (open source licence, https:/imagej.nih.gov—accessed on 3 May 2021) was used to measure the lesion area. A periodontal probe with a millimetre scale, applied next to the treated area, served as a mapping scale for calculating the lesion area. Calculations were performed before treatment and at 3 months after treatment in cases where no or an incomplete response was observed. The mean area of the treated lesions was 38 mm^2^ (SD = 48). The smallest lesion subjected to treatment was 10 mm^2^, and the largest was 220 mm^2^.

### 2.5. Fractal Dimension Analysis

Fractal dimension analysis was performed in each case before lesion treatment as well as at 7 days and at 3 months after treatment. The control group consisted of normal mucosa. ROIs of reference mucosa were taken 3 months after surgery from the same location as the examined lesion. The ROI was dependent on the size of the lesion. Three ROIs were used in the study: 50 × 50, 100 × 100 and 150 × 150 pixels, with images taken 3 months after surgery from the same location as the lesion; e.g., if the lesion was in the control lip, the ROI was taken from the healthy lip area. All fractal analysis was performed in ImageJ version 1.53e (Image Processing and Analysis in Java—Wayne Rasband and contributors, National Institutes of Health, USA, public domain license, https://imagej.nih.gov/ij/, accessed on 3 May 2021) and plugin FracLac version 2.5 (Charles Sturt University, Australia, public domain license).

In our research, we decided to use a modified box counting method algorithm that allowed for the analysis of 8- or 16-bit monochrome images. For greyscale images, FracLac offers three options for FD (fractal dimension) analysis. One of these options is intensity difference, which we applied in our study. The analysed image was divided into boxes as in the box counting method. The difference between the maximum pixel intensity and the minimum pixel intensity was counted in each box (δI_i,j,ε_, where i, j denotes the position of the analysed box in the scale ε):δI_i,j,ε_ = maximum pixel intensity_i,j,ε_ − minimum pixel intensity_i,j,ε_

In the next step, 1 is added to the intensity difference to prevent 0 value:I_i,j,ε_ = δI_i,j,ε_ + 1

Finally, the fractal dimension of the intensity difference is described by the following formula:$$D\;\mathrm{Idiff}=\underset{\varepsilon \to 0}{\lim}\frac{\ln\left(I\varepsilon \right)}{\ln\left(\frac{1}{\varepsilon }\right)}$$
where D Idiff—difference intensity fractal dimension, I_ε_ = Σ[1δI_i,j,ε_ +1] and ε—box scale.

All operations are shown in Figure 1.

### 2.6. Texture Analysis

The texture of oral mucosa was evaluated using features derived from a co-occurrence matrix [19,20]. The regions of interest (ROIs) were normalized (μ ± 3σ) to share the same average (μ) and standard deviation (σ) of optical density within the ROIs. Selected image texture features (entropy and difference entropy (DifEntr) from the co-occurrence matrix; a good short- and long-run emphasis moment from the run length matrix) in ROIs were calculated for each composite material tested:(1)$$\mathrm{DifEntr}=-\sum_{i=1}^{Ng}p_{x-y}\left(i\right)log\left(p_{x-y}\left(i\right)\right)$$
(2)$$\mathrm{Contrast}=\overset{Ng-1}{\underset{\left|i-j\right|=0}{\sum }}{\left(\left|i-j\right|\right)}^{2}\overset{Ng}{\underset{i=1}{\sum }}\overset{Ng}{\underset{j=1}{\sum }}p\left(i,j\right)$$
where Σ is the sum, Ng is the number of optical density levels in the radiograph, i and j are the optical density of pixels that are 5 pixels away from one another, p is probability and log is the common logarithm. The equation was a measure of microcontrast because the image was sampled at distances of 5 pixels.

### 2.7. Statistical Analysis

Statistica version 13.3 (StatSoft, Krakow, Poland) was used to perform all statistical tests. A statistical significance level of 0.05 was assumed. The Shapiro–Wilk test was applied to confirm the normality of the distribution. Parametric tests were carried out due to the normality of the distribution. Analysis of variance (ANOVA) and post hoc least significant difference were used to show differences in fractal dimensions between normal mucosa and lesions before treatment as well as at 7 days and at 3 months after treatment. A correlation matrix was used to assess the FD correlation between the measured area and the fractal dimension before and after lesion treatment.

We calculated the power of all applied statistical tests on the basis of sample size (n = 23). At a level of alpha of 0.05, the powers of tests in these conditions were 0.71 (for difference entropy), 0.88 (FD) and 0.94 (microcontrast).

## 3. Results

A complete therapeutic effect without scar formation was achieved in 83% (19 lesions). In 9% (two cases), scar formation was observed after three months. No healing was observed in 4% (1 case) and partial healing (reduction in lesion size) in 4% (1 case). An example of total response treatment is shown on Figure 2.

The overall results of the surface area and fractal dimension values are shown in Table 1. The mean lesion size was 38 mm^2^; it was significant that the standard deviation was very high (48 mm^2^). The smallest lesion had an area of 10 mm^2^, and it was located in the cheek region, and the largest lesion had an area of 220 mm^2^ and was located in the lip region. The fractal dimension (FD) was smaller for the lesion before treatment (1.359) and 1 week after treatment (1.354) than for the lesion 3 months after treatment (1.440). The results of the post hoc ANOVA (least significant difference) test are shown in Table 2. There were no statistical differences between FDs of lesions before and 7 days after laser treatment. It should be noted that our results showed statistical differences between the changes in FDs before treatment and 3 months after treatment. No statistically significant differences were observed between the lesion after three months and the healthy mucosa, confirming that fractal dimension analysis was a useful method to objectively assess the treatment effect.

The results did not confirm the linear correlation between the area and the fractal dimension of the lesion before treatment (Pearson coefficient, r = 0.154).

Digital texture analysis is shown in Figure 3. The treated vascular lesions resulted in decreased colour contrasts in the oral mucosa (*p* < 0.001). Both the lesion image and the condition 7 days after treatment have reduced contrast. As a result of the treatment (after 3 months), the texture of the mucosa image was restored to normal (Table 3 and Figure 4). When examining difference entropy as a measure of random patterns in texture, no change in the treatment process was observed.

The FD was moderately strongly related with texture microcontrast in images of the pathological lesion (Figure 5), the 3-month healing effect and of healthy oral mucosa (before treatment: R2 = 26%, correlation coefficient = 0.51, *p* < 0.05; 3 months after: R2 = 30%, correlation coefficient = 0.55, *p* < 0.01; control: R2 = 27%, correlation coefficient = 0.52, *p* < 0.05). The image of the healing wound on the seventh postoperative day alone did not reveal an association of the FD with the contrast (R2 = 2.7%, CC = 0.16, *p* = 0.45).

## 4. Discussion

VL is most often not accompanied by distressing symptoms and is usually not a problem for patients. This is why few of them see a doctor regarding this condition. Aesthetic reasons or possible bleeding that may result from mechanical trauma to these lesions are two of the main factors that lead patients to seek help [6]. It is therefore difficult to establish the prevalence of VL in the population. Meeni et al., in their study, stated that the mean age of patients treated for VL was 76.7 years [21], Cebeci et al. reported 62.1 years [22] and Azewedo et al., 55 years [23]. In our study, the mean age was 66 years, which confirmed the literature stating that these types of lesions appear at an older age. Gender predilections are also difficult to establish. Voynov et al. [24] indicated a higher prevalence of VL in men with an M:F ratio of 4:1, Menni et al. [21] reported M:F—1.48:1 while Azewedo et al. [23] reported M:F—0.7:1. Our findings showed that the frequency of VL was similar in both sexes with a slight female predominance and M:F—0.91:1.

800–980 nm diode lasers are often used to treat superficial vascular lesions in the oral area, as they are a safe and effective therapeutic method [23,24,25]. The non-contact application of the 980 nm diode laser is highly effective in treating these lesions due to the high absorption of this wavelength by haemoglobin, whose absorption spectra are between 410 and 600 nm and 800 nm and 1100 nm [9]. As a result of this process, the energy of the laser beam is converted into heat and causes coagulation of proteins and obliteration of pathological blood vessels. In addition, the heat generated during absorption of the laser light energy causes photocoagulation of well-vascularised tissues to a depth of 7–10 mm [6]. This makes this type of device very suitable for treating vascular surface lesions. Our results are consistent with the literature, which reports that the transmucosal application of lasers is associated with a low incidence of complications such as bleeding and oedema and is characterised by low peri- and postoperative pain [26,27,28]. The non-contact application of lasers also includes a spot irradiation technique called the “Leopard technique”, which involves irradiating the vascular lesion with a distance of several millimetres between the irradiated areas. This prevents excessive heat accumulation and minimises epithelial damage, resulting in better healing. In addition, the occurrence of complications such as pain, excessive swelling, ulceration or scarring is reduced. However, this type of approach may require multiple treatment sessions, as confirmed by the study of Miyazaki et al., who used this technique as well as the Nd:YAG—1064 nm laser (VersaPulse Holmium/Nd-YAG Laser Dual Wavelength model; Lumenis, Ltd., Yokneam, Israel) to treat venous malformation in the oral cavity [29]. This mode of irradiation can be used alone or in combination with other techniques, especially in cases of large or deeper-lying vascular lesions.

Established therapeutic approaches for VL include, among others, photocoagulation using various laser devices such as a diode laser (800 nm–980 nm), a CO2 laser—10,800 nm, a neodymium-doped yttrium aluminium garnet laser (Nd:YAG) (1064 nm), an alexandrite laser (755 nm) or a potassium titanyl phosphate laser (KTP)—532 nm [23,30,31,32,33].

The study of Nammour et al., using four laser devices including a CO2 laser (10.600 nm) (Smart US20 D Laser, High Tech Laser, Herzele, Belgium), a continuous wave (CW) at 1 W power, a diode laser at 980 nm (Smart M Pro, Lasotronix, Poland), a 4 W, CV, ND:YAG 1064 nm (Fidelus Plus, Fotona Medical Laser, Ljubljana, Slovenia) in pulsed mode (PM) at 15 Hz and an Er,Cr:YSGG 2790 nm (Waterlase, Biolase, Inc., Foothill Ranch, CA, USA) at 0.25 W and 20 Hz for the treatment of vascular lesions in the oral area, confirming their high efficacy regardless of the wavelength or treatment procedure used [34]. They reported no significant differences in aesthetic healing 6 months after treatment. No recurrences were observed with CO_2_ and Er,Cr:YSGG lasers as opposed to diode and Nd:YAG lasers, which were 11% and 0.9%, respectively. However, in the case of the Er,Cr:YSGG laser and the CO_2_ laser, invasive methods consisting of excision and vaporisation of the lesion were also used. The other two lasers were minimally invasive and non-contact methods such as in our study. Our observations showed that we obtained a similar result in cases of absent or incomplete healing of lesions, which were only 8% among all cases.

Voynov et al., conducting a study using a 980 nm diode laser (LiteMed-ics^®^, Italy), 2–3 W, CW, in non-contact mode, reported that they achieved complete healing of the VL in all treated—35 patients over a period of 2–4 weeks [24]. Furthermore, in no case did they observe scar formation, hypopigmentation or hyperpigmentation or mucosal atrophy, which was in contradiction with our results. Similar results were reported by Azevedo et al. using an 808 nm diode laser (Lasering 808, Milan, Italy), 2–3 W, CV, in non-contact mode for the treatment of VL, where they obtained a complete response in all 17 cases over a period of 2–3 weeks [23]. It took only one therapy session to achieve this result. Different results were reported by Bacci et al., who conducted a study on VL and venous malformation using an 830 nm diode laser (Opus 10, Sharplan Laser Industries, Ltd., Tel-Aviv, Israel), 1.6 W, in continuous mode [35]. After 30 days, complete healing was achieved in 74.5% and after one year, in 81.3%. Only 6.7% of patients required a repeat therapy session and 3.3%, three sessions. Their results were in line with our observations, where in 4% of cases, no improvement was obtained, and one patient required an additional therapy session, also representing 4% of the subjects. Our results were similar in that 4% needed another therapy session.

Romeo et al. conducted a study using a KTP 532 nm (DEKA, Florence, Italy), 2.5 W, CV laser and a GaAlAs diode laser (Laser Innovation, Castelgandolfo, Italy, 808 nm) to treat benign vascular lesions in the oral area [27]. With the KTP laser, three techniques were used: irradiation using a non-contact technique applying a microscope slide to the mucosa, irradiation by placing an optical fibre inside the lesion and using the laser to cut out lesions. The 808 nm diode laser was only used to irradiate the inside of the lesion. In all 13 cases, they reported complete healing, and only 1 patient required a repeat procedure. Only superficial anaesthesia with lidocaine cream was used for a period of 10 min before the procedure itself. Among all the methods used, patients reported an intraoperative NRS score of −1.92 and in the case of transscleral photocoagulation using a basic slide, 1.77. In our study, the intraoperative NRS score was 4.76. Therefore, non-contact VL treatment can be considered minimally invasive and safe, accompanied by moderate pain.

The application of no-contact mode through a transparent microscope slide reduces the vertical dimension of lesion and enables better penetration of the laser beam deep into the tissue. This procedure increases light penetration, but still, the main limitation of any laser procedure is the depth of beam penetration.

In previous studies, we applied the FDA when analysing the efficacy of oral leukoplakia treatment. In that study, we used the classical box counting method to estimate the count up fractal dimension. This method has one major drawback, in that it requires 1-bit images as source material for analysis. Converting a colour image to a single-bit image is a process in which some of the analysed details are lost. We showed that the FD of oral leukoplakia was significantly lower than that of healthy mucosa with no statistical differences between the treated lesion and healthy mucosa [36]. In the aforementioned study, the FD of the examined lesion was lower than that of the reference mucosa, and after treatment, it was close to that of the normal mucosa, as in our study. These results confirmed our previous observations that FD was a useful method to objectively describe post-treatment lesions compared to healthy mucosa. Lucchese et al. investigated the fractal dimension of the capillary pattern of the oral lichen planus. They showed that the erosive form of oral lichen planus had a higher FD value (1.167) compared to healthy control mucosa (1.123) [37]. Iqbal et al. demonstrated that the fractal dimension of oral leukoplakia with dysplastic lesions was significantly greater than that of non-dysplastic lesions [38]. Goutzanis et al. studied the vascular pattern of oral cancer in histological sections. Their study showed that cancer vessels had a higher FD value than normal mucosa [13].

Although DifEntr is a frequently used texture feature in medical image recognition, it did not perform well in observing the healing process of vascular lesions (*p* = 0.1773). It is reasonable to assume that this was due to the structured shape of the vascular lesions and the low stochastic components in their images (low measures of chaos such as entropy in texture) [39,40,41,42,43,44,45].

On the other hand, the microcontrast of the texture (variable: contrast) presented in the photographic images in this study gave full agreement with clinical observation and fractal analysis (*p* < 0.05). It can be speculated that relatively large venous lacunae significantly affect the image structure. They lower the overall ROI contrast, which increases with healing. Finally, the microcontrast of the treated site reaches the level of normal oral mucosa. The fit of this feature to the lesions studied here is not surprising. Similar features derived from the co-occurrence matrix (sum of average) have already proven successful in oral surgery [46,47,48]. The contrast feature analysis of the structure recorded in the vascular lesion photographs used in this study revealed microstructures of keratinized epithelium. This was due to good visibility of the whitish structures of the superficial layer of the epithelium. They increased the value of the microcontrast parameter. The anatomically/histologically structured epithelial lining is also important. This stroma in a vascular lesion is simply the contents of the vascular malformation contrary to control sites, where the epithelium lies on collagen-structured connective tissue. The authors see in these conditions an explanation for the obtained results of the texture analysis.

## 5. Conclusions

Fractal dimension and texture analysis was a useful and objective method for assessing the treatment effects on venous lake lesions treated with diode lasers;The fractal dimension and microcontrast of a vascular lesion were mutually coupled;Fractal dimensions of the venous lakes were significantly lower than healthy mucous. There was no statistical difference between the FDs of healthy mucous and the lesions 3 months after treatment.The venous lake lesions had significantly lower contrast than normal mucosa (similar site conditions 7 days after treatment). After 3 months, the treated site achieved texture features that were the same as the intact mucosa.The non-contact mode of the 980 nm diode laser was an effective and safe method of treating a venous lake.

## Figures and Tables

**Figure 1 materials-14-04140-f001:**
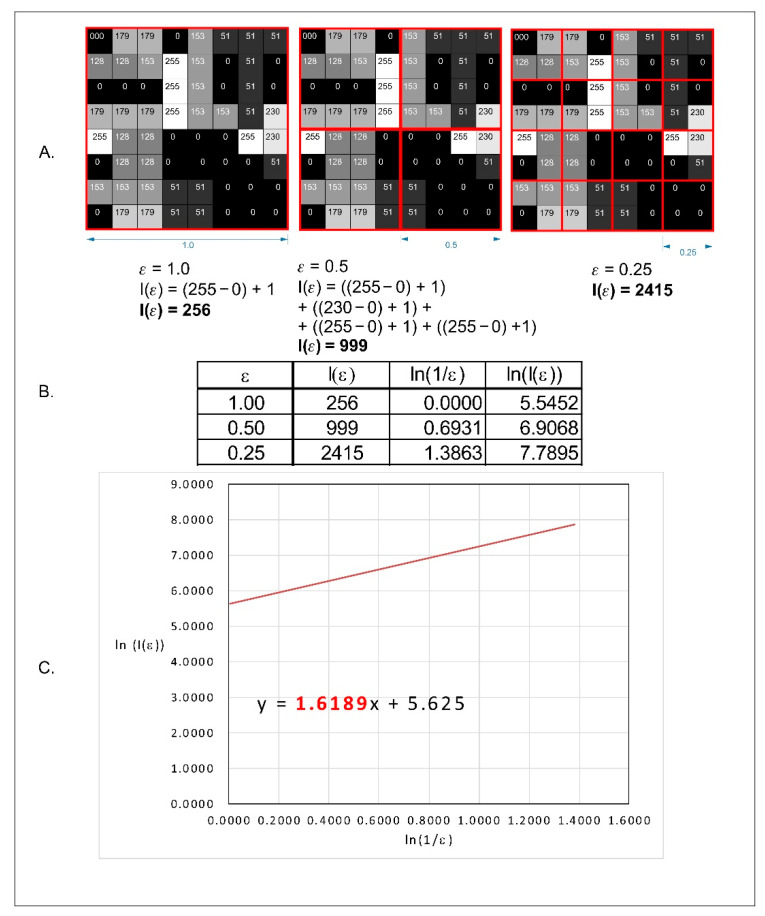
Graphical interpretation of the intensity difference algorithm of the fractal dimension calculation. (**A**) An example of a greyscale 8-bit image; the numbers in squares represent the intensity level of each pixel: 0, black and 255, white. The red squares represent the scale—ε. (**B**) The values of the intensity difference for each step of scale reduction (ε). (**C**) A straight line drawn through the points from table B on the x–y chart in a natural logarithmic scale. The slope factor of this straight line is a value fractal dimension counted by the intensity difference algorithm.

**Figure 2 materials-14-04140-f002:**
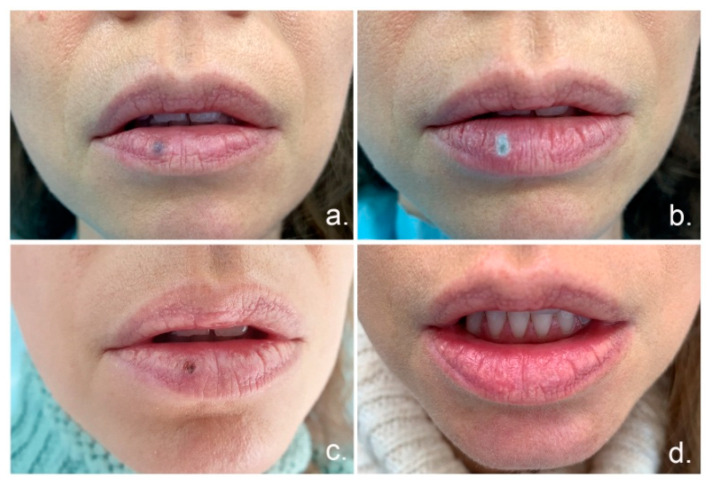
Application of a diode laser, 980 nm, in the treatment of a venous lake of the lower lip. (**a**) The appearance of lesions before treatment, (**b**) the appearance immediately after applying laser application, (**c**) 7 days after treatment (**d**) 3 months after treatment.

**Figure 3 materials-14-04140-f003:**
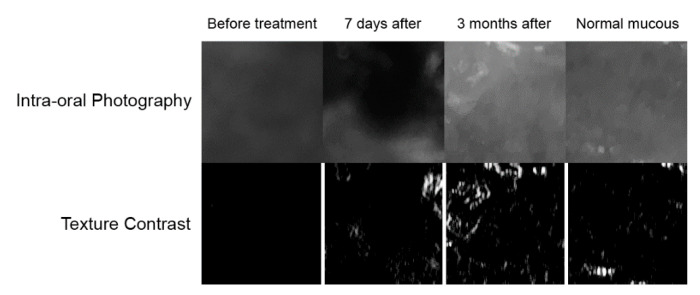
Digital texture analysis: The top row of images represents treatment documentation captured with a digital camera. These are black and white photographs. Below the bottom row of images are presented maps of intensity of the texture feature under study. In white in these maps are marked areas of high microcontrast detection. Conversely, black areas indicate low intensity of the contrast feature. Healing is characterized by the appearance of scattered areas of high microcontrast.

**Figure 4 materials-14-04140-f004:**
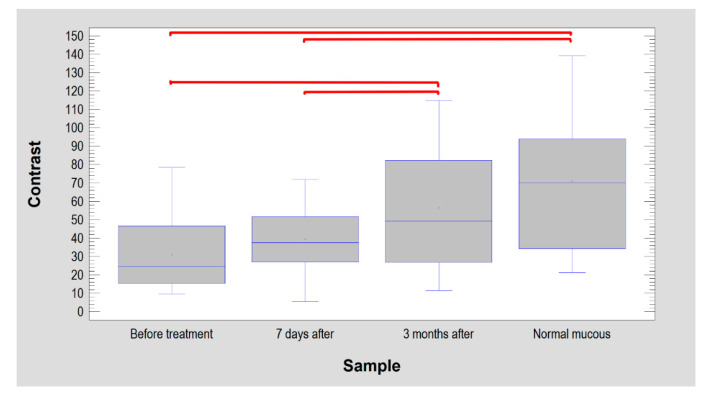
Texture contrast findings in oral mucosal images. The lesion had significantly lower contrast than normal mucosa (similar site condition 7 days after treatment). After 3 months, the treated site achieved texture features that were the same as the intact mucosa. The horizontal lines in the boxes indicate the median. Red brackets show groups with statistically significant differences (*p* < 0.0001).

**Figure 5 materials-14-04140-f005:**
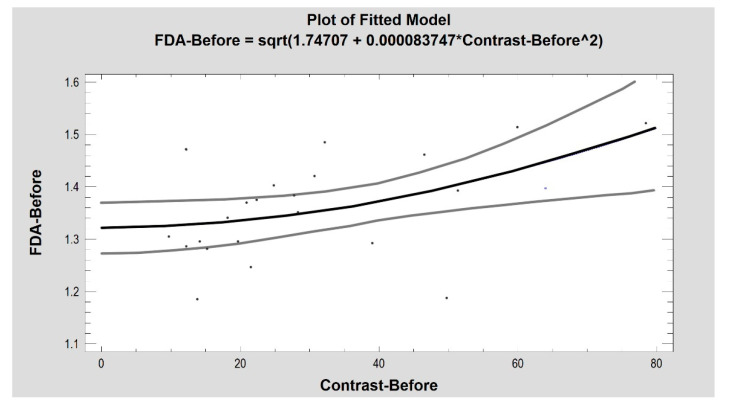
The fractal dimension and microcontrast of a vascular lesion are mutually coupled (*p* < 0.05). Grey lines indicate confidence limits.

**Table 1 materials-14-04140-t001:** Mean values and standard deviations (SD) of lesion areas and fractal dimension (FD) values before treatment, 7 days after treatment, 3 months after treatment and with normal mucosa.

Number of Lesions		Surface (mm^2^)	FD Value			
			Before Treatment	7 Days After	3 Months After	Normal Mucosa
**23**	Mean	38	1.359	1.354	1.440	1.500
	SD	48	0.095	0.087	0.155	0.135

**Table 2 materials-14-04140-t002:** Post hoc ANOVA results (least significant difference) for comparison of FD values of lesions before treatment, lesions at 7 days and at 3 months after treatment and normal (healthy) mucosa. (bold and underline font—statistically significant differences, *p* < 0.05).

vs.		FD Value			
		Before Treatment	7 Days After	3 Months After	Normal Mucosa
**FD value**	Before treatment		0.892209	**0.026944**	**0.000162**
	7 days after	0.892209		**0.019182**	**0.000099**
	3 months after	**0.026944**	**0.019182**		0.094181
	Normal mucosa	**0.000162**	**0.000099**	0.094181	

**Table 3 materials-14-04140-t003:** Mean values of microcontrast and difference entropy of normal mucous, lesions before treatment, 7 days after treatment and 3 months after treatment.

Texture Feature	Before Treatment	7 Days After	3 Months After	Normal Mucous	Note
Microcontrast	31 ± 19	40 ± 18	57 ± 32	71 ± 35	*p* < 0.0001
Difference Entropy	1.00 ± 0.11	1.04 ± 0.15	1.08 ± 0.12	1.03 ± 0.15	*p* = 0.1773

## Data Availability

Data are available from the authors at mateusz.trafalski@umed.wroc.pl, kamil.jurczyszyn@umed.wroc.pl.

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
