# Peer review of "Application of Fractal Dimension and Texture Analysis to Evaluate the Effectiveness of Treatment of a Venous Lake in the Oral Mucosa Using a 980 nm Diode Laser—A Preliminary Study"

_materials, 2021, doi:10.3390/ma14154140_

Round 1

Reviewer 1 Report

  Thank you for submitting the paper   Concerns: Language needs improvement, especially in the Introduction and the Discussion.    Abstract: the authors are suggested to provide a bit more detailed information in method and result parts. Add Statistical analysis in the Material and methods section.)   Introduction What is the importance of this study? You do not think this study is included to the others already done? Which results are comparable? What this study has new?   Materials and methods. Sample size´s calculation? Has it not been possible to compare it with a control group with a gold standard technique?

Discussion: Study limitations?

Author Response

Dear Reviewer,

We would like to thank You for Your valuable suggestions to improve our study. We performed following changes:

Languages has been improved, we attach professional translation certificate.

Abstract limitation is 200 words. We tried to nor exceed this limit. We added power of laser in methods section. In results we added:  The fractal dimension and microcontrast of a vascular lesion are mutually coupled”.

All statistical methods are described in chapter 2.7.

We did not find in literature more studies about application of fractal dimension and texture analysis in case of treatment evaluation of venous lake lesions.  So it is an main novelity aspect of our study (We added this information to the introduction). The first aim of the study was to investigate the efficacy of oral mucous venous lake lesions treatment using a 980nm diode laser. The second goal of the study was the application of texture and fractal dimension analysis in the assessment of the effectiveness of treatment of these lesions.

In most of other studies only lesion size is analyzed before and after treatment and it is an only one result which could be comparable.

We calculated power of all applied statistical tests on the base of sample size (n=23). At the level of alfa as 0.05, power of tests in these conditions was 0.71 (for Difference Entropy, in this case alternate hypothesis was not confirmed), 0.88 (FD) and 0.94 (microcontrast). We added this information to the chapter 2.7.

There is no golden standard set for treatment of venous lake lesions. Surgical excision, cryosurgery using liquid nitrogen, electrocoagulation, coagulation with infrared light, photocoagulation and sclerotherapy are widely used in clinical practice. It is preliminary study (we added it to the title).  In our future study we will compare near infrared photocoagulation with other wavelengths of visible spectrum because we confirmed in our previously study synergistic effect of 460 and and 540 nm wavelengths in thermal effect [Kamil Jurczyszyn, Witold Trzeciakowski, Zdzisław Woźniak, Piotr Ziółkowski, Mateusz Trafalski.Assessment of Effects of Laser Light Combining Three Wavelengths (450, 520 and 640 nm) on Temperature Increase and Depth of Tissue Lesions in an Ex Vivo Study. Materials 2020, 13(23), 5340; https://doi.org/10.3390/ma13235340].

Application of no-contact mode through a 1mm thick transparent microscope slide reduces vertical dimension and thus better penetrate the laser beam deep into the tissue. This procedure increase a light penetration but still main limitation of any  laser procedure is a depth of penetration.  We added this information to the discussion section.

Best regards,

Authors.

Reviewer 2 Report

Manuscript is well written and well described in a scientific manner however I have few suggestions.

  1. Introduction is too lengthy and too boring. It should be very concise and to the point. I will also suggest to reduce the length to one and half page only.
  2. Figure 5 is too dull. Please revamp the image.
  3. Conclusion is too short. It should conclude the studies outcome in a very precise and scientific way.
  4. Digital texture analysis image also needs improvement. 

Author Response

Dear Reviewer,

We would like to thank You for Your valuable suggestions to improve our study. We performed following changes:

Introduction is too lengthy and too boring. It should be very concise and to the point. I will also suggest to reduce the length to one and half page only.

We reduced length of introduction by removing second and third paragraph.

Figure 5 is too dull. Please revamp the image.

As for the relationship between FD and TA, the relationship is not known. Therefore, it is useful to show how the two parameters depend on each other. Both features can be called second-order features because they do not come directly from the image appearance. On the contrary, they arise as much processed derivatives of the relation hidden in photographic images. The dependence of FD on microcontrast has never been detected in other world publications. Or perhaps it has never been studied in other studies. Therefore, it is worthwhile to disseminate this unique dependence and point it to other scientific teams for further investigation.

We modified the figure to be more detailed.

Conclusion is too short. It should conclude the studies outcome in a very precise and scientific way.

We added following conclusions:

  1. Fractal dimension and texture analysis is a useful and objective method of assessing the treatment effect of venous lake lesions treated with diode laser.
  2. The fractal dimension and microcontrast of a vascular lesion are mutually coupled.
  3. Fractal dimension of venous lake is significant lower than healthy mucous. It is no statistical difference between FD of healthy mucous and lesion 3 months after treatment.
  4. The venous lake lesions have significantly lower contrast than normal mucosa (similar site condition 7 days after treatment). After 3 months, the treated site achieves texture features the same as the intact mucosa.
  5. The non-contact mode of the 980 nm diode laser is an effective and safe method of treating a venous lake.

Digital texture analysis image also needs improvement.

We added to the discussion:

The contrast feature analysis of the structure recorded in vascular lesion photographs, used in this study, reveals microstructures of keratinized epithelium. This is due to the fact, good visibility of the whitish structures of the superficial layer i epithelium. They increase the value of the microcontrast parameter. The anatomically/histologically structured epithelial lining is also important. This stroma in a vascular lesion is simply the contents of the vascular malformation. Contrary to control sites, where the epithelium lies on collagen-structured connective tissue. The authors see in these conditions an explanation for the obtained results of the texture analysis.

Best regards,

Authors.

Round 2

Reviewer 2 Report

The changes are included and now it is improved and simply can be punlished